# Replication of bacterial plasmids in the nucleus of the red alga *Porphyridium purpureum*

Zhichao Li[1] & Ralph Bock 🄳 [1]

Rhodophytes (red algae) are a diverse group of algae with great ecological and economic importance. However, tools for post-genomic research on red algae are still largely lacking. Here, we report the development of an efficient genetic transformation system for the model rhodophyte *Porphyridium purpureum*. We show that transgenes can be expressed to unprecedented levels of up to 5% of the total soluble protein. Surprisingly, the transgenic DNA is maintained episomally, as extrachromosomal high-copy number plasmid. The bacterial replication origin confers replication in the algal nucleus, thus providing an intriguing example of a prokaryotic replication origin functioning in a eukaryotic system. The extended presence of bacterial episomal elements may provide an evolutionary explanation for the frequent natural occurrence of extrachromosomal plasmids in red algae, and may also have contributed to the high rate of horizontal gene transfer from bacteria to the nuclear genome of *Porphyridium purpureum* and other rhodophytes.

[1] Max-Planck-Institut für Molekulare Pflanzenphysiologie, Am Mühlenberg 1, D-14476 Potsdam-Golm, Germany. Correspondence and requests for materials should be addressed to R.B. (email: rbock@mpimp-golm.mpg.de)

The red algae (Rhodophyta) represent one of the oldest and largest groups of eukaryotic algae. They occur in fresh-water and marine environments and show great morphological diversity, covering both unicellular and complex multicellular species. Their red color comes from phycobiliproteins in the plastids which serve as light-harvesting pigment–protein complexes in photosynthesis. The plastids of red algae go back to the ancestral primary endosymbiosis that gave rise to photosynthetic eukaryotes, and thus are surrounded by only two membranes. Together with green algae, plants and glaucophytes, red algae form the (likely monophyletic) group Archaeplastida. During evolution, the red algal plastids spread through secondary endosymbioses into several other lineages of photosynthetic eukaryotes, including diatoms, dinoflagellates, and haptophytes[1,2].

Despite their diversity and their ecological importance—plastids of red algal origin account for most of the carbon fixed in the world's oceans—red algae remain an understudied group. A few red algae are on the way to becoming model organisms, including the thermoacidophilic species *Cyanidioschyzon merolae* and *Galdieria sulphuraria*, the red seaweed *Chondrus crispus*, and the mesophilic rhodophyte *Porphyridium purpureum*. Their genomes have been sequenced and some tools for molecular research in these species have been developed, most notably a genetic transformation system for *Cyanidioschyzon merolae* that seems to exhibit a relatively high rate of homologous recombination[3]. However, the development of genetic tools for red algal research is still in its infancy, and currently most species remain non-transformable[4].

Here we have sought to address this bottleneck by developing a workable genetic transformation system for the red alga *P. purpureum*. *P. purpureum* is a unicellular alga that is widespread in marine and terrestrial ecosystems and can be grown in artificial seawater in the laboratory. Its genome has been sequenced[5], and one of the most remarkable findings from the genome project has been the discovery of hundreds of genes originating from horizontal gene transfer (HGT) events. Out of the 8355 predicted genes, 5.4–9.3% show evidence of a reticulated evolutionary history that involved HGT and/or endosymbiotic gene transfer, with bacteria being the main source organisms of HGT[5]. Why *P. purpureum* is such an efficient recipient of genes that are horizontally transferred from bacteria is currently unknown.

Exploring methods to introduce foreign DNA into the nucleus of *P. purpureum*, we report here an efficient transformation system and show that reporter proteins can be expressed to very high levels and targeted to distinct subcellular destinations. Surprisingly, the introduced transformation vectors are episomally maintained and stably replicated in the nucleus of *P. purpureum*. Replication of bacterial genetic elements inside a eukaryotic compartment raises intriguing questions about algal genome evolution and HGT. It may explain the frequent presence of plasmid-like elements in red algal genomes[6,7] and may also have relevance to the high abundance of genes of bacterial origin in the nuclear genome of *P. purpureum*.

## Results

### Development of a transformation method for *P. purpureum*.
To identify selectable marker genes that are potentially suitable for the development of a genetic transformation system for *P. purpureum*, the sensitivity of algal cells to possible selection agents was determined. Tests of various antibiotics for their potential to inhibit algal growth on agar-solidified culture media (Fig. 1a; Supplementary Fig. 1) revealed a remarkable tolerance of *P. purpureum* to most antibiotics that efficiently inhibit growth of other algae and embryophyte plants (Supplementary Fig. 1). However, three antibiotics were found to efficiently suppress algal growth: the chloroplast translational inhibitor chloramphenicol, and the two DNA strand break-inducing drugs zeocin and bleocin (both belonging to the bleomycin family of antibiotics; Fig. 1a; Supplementary Fig. 1).

We then constructed expression cassettes based on suitable antibiotic resistance genes. Chloramphenicol resistance is conferred by the *cat* gene from *Escherichia coli* which encodes the antibiotic-inactivating enzyme chloramphenicol acetyltransferase. Resistance to zeocin and bleocin is conferred by the *ble* gene from the actinomycete bacterium *Streptoalloteichus hindustanus*. The gene product of *ble* is a small protein that binds bleomycin antibiotics in a 1:1 stoichiometry, and, in this way, prevents them from intercalating into DNA and inducing strand breaks. Both resistance genes are widely used as selectable marker genes in prokaryotic and eukaryotic systems, including eukaryotic algae[8–12]. To drive selectable marker gene expression in the nucleus of *P. purpureum*, an expression cassette derived from the promoter, 5′ untranslated region and terminator region of a constitutively expressed housekeeping gene of *P. purpureum*, the α-tubulin gene, was constructed (Fig. 1b). In addition, standard expression signals for nuclear transgene expression in seed plants (the CaMV 35S promoter and the *nos* terminator) were also tested for their suitability to drive selectable marker gene expression.

Large-scale transformation experiments were conducted using two different methods: electroporation and particle gun-mediated (biolistic) transformation. While electroporation experiments did not result in antibiotic-resistant colonies, biolistic transformation with the *ble* marker driven by tubulin-derived expression signals and subsequent selection for zeocin resistance produced algal clones that grew in the presence of the antibiotic (Fig. 1b). By contrast, the *ble* expression cassette based on the CaMV 35S promoter and *nos* terminator did not produce resistant colonies, suggesting that these expression signals do not function in *P. purpureum*. Also, selection for resistance to bleocin or chloramphenicol did not give resistant colonies. Therefore, biolistic transformation in combination with selection for zeocin (and the *ble* marker driven by tubulin expression signals) was used in all subsequent experiments.

### Expression of fluorescent reporter proteins in *P. purpureum*.
As spontaneous resistance to zeocin is not known to occur, we preliminarily assumed that the obtained zeocin-resistant colonies were transgenic. To directly visualize transgene expression, we constructed vectors that contain the gene for the green fluorescent protein (GFP) as a reporter of gene expression whose accumulation levels can be readily determined by either measuring fluorescence or determining protein levels with the help of sensitive antibodies. In one vector (pZL19; Fig. 1b), the *GFP* coding region was fused to the *ble* gene resulting in expression of a fusion protein that should localize to the cytosol and the nucleus. In a second vector (pZL22; Fig. 1b), *GFP* was expressed from a separate expression cassette constructed based on expression signals of the actin gene of *P. purpureum*. To also examine whether the fluorescent reporter can be targeted to a different subcellular compartment, the *GFP* in pZL22 was fused to a transit peptide for protein import into the plastid. The transit peptide sequence was taken from the *P. purpureum ATPC* gene encoding the γ-subunit of the plastid ATP synthase in the thylakoid membrane.

When introduced into *P. purpureum* cells by biolistic bombardment, the selected zeocin-resistant clones displayed strong green fluorescence that showed the expected subcellular localization (Fig. 1c). Strains obtained with plasmid pZL19 (*Pp*-ZL19 clones) showed strong GFP accumulation in the nucleus and the cytosol, whereas strains generated with vector pZL22 (*Pp*-ZL22 clones) displayed strong GFP labeling of the plastid (as

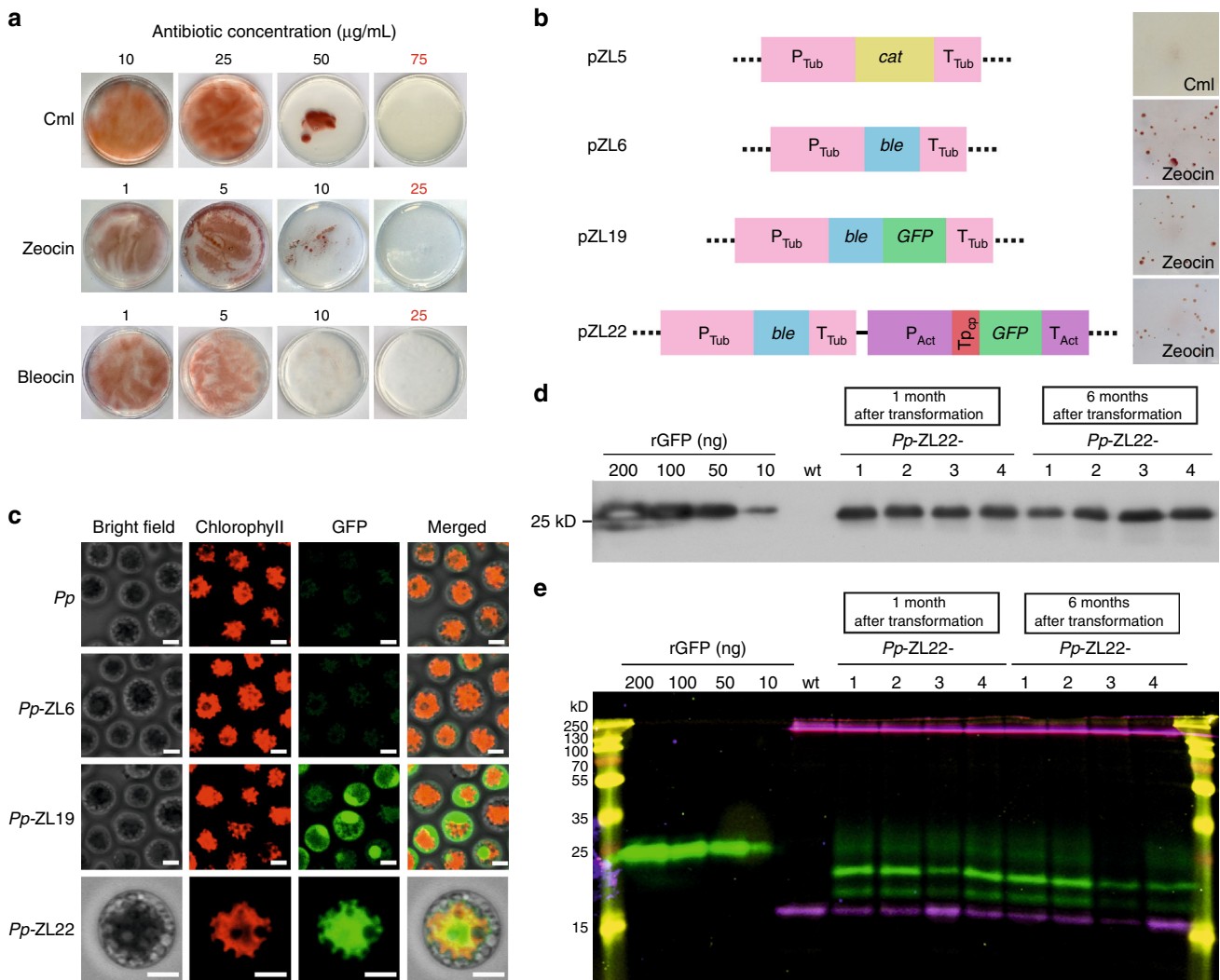

**Fig. 1** Transformation of *Porphyridium purpureum*, targeting of GFP to different subcellular locations, and analysis of GFP accumulation in transgenic algal strains. **a** Antibiotic sensitivity tests for *P. purpureum* on ASW-containing agar-solidified medium. The algal cells are sensitive to the plastid translational inhibitor chloramphenicol (Cml) and the DNA-cleaving agents zeocin and bleocin. Red numbers denote the antibiotic concentrations used for selection of transformed cells. For tests of additional antibiotics and additional concentrations, see Supplementary Fig. 1. **b** Schematic representations of the expression constructs used for transformation and identification of transgenic colonies on selective medium. P promoter, T terminator, Tp transit peptide for protein import in plastids. **c** Analysis of GFP accumulation in transgenic red algae by confocal laser-scanning microscopy. Scale bars: 2.5 μm. **d** Immunoblot analysis of GFP expression in red algae transformed with construct pZL22. 1 μg total soluble protein was loaded in each lane. A dilution series of recombinant GFP (rGFP) was included for semiquantitative assessment of GFP accumulation. Protein accumulation is estimated to be approximately 5% of the total soluble protein (TSP). Protein accumulation levels were determined 1 and 6 months after transformation. The strains were grown in zeocin-containing medium. Wt wild type. **e** In-gel fluorescence analysis of GFP expression. Samples of 1 μg TSP extracted from transgenic algae transformed with plasmid pZL22 and a dilution series of rGFP were analyzed under non-denaturing electrophoresis conditions[13] followed by fluorescence imaging. Shown is a merged image of three images taken with three different excitation wavelengths. GFP (green bands) was visualized by excitation at 488 nm, phycoerythrin (purple bands) was visualized at 532 nm, and the marker bands (yellow) at 633 nm. The individual images can be seen in Supplementary Fig. 2a

revealed by the characteristic morphology of the single star-like plastid present in each algal cell and the overlay with the chlorophyll fluorescence). Algae transformed with the control vector pZL6 (*Pp*-ZL6 clones) that does not contain the *GFP* transgene showed no above-background fluorescence (Fig. 1c). GFP levels were remarkably similar in all transgenic algal clones analyzed, as evidenced by both immunoblotting (Fig. 1d) and in-gel fluorescence assays[13] (Fig. 1e; Supplementary Fig. 2a). When protein accumulation levels were quantified using a dilution series of purified GFP recombinantly expressed in *E. coli*, GFP was estimated to accumulate to approximately 5% of the total soluble protein (TSP). This level is substantially higher than what can be achieved even with fully optimized expression constructs

and specific expression strains in the model alga *Chlamydomonas reinhardtii*[13–15].

Transgene expression was stable over time, and when GFP expression was reanalyzed half a year after transformation, no decrease in protein accumulation levels was observed (Fig. 1d, e; Supplementary Fig. 2a), indicating that no transgene silencing occurs.

**Episomal maintenance of transgenes in *P. purpureum*.** To characterize the transgenic loci in transformed *P. purpureum* clones, Southern blot experiments were conducted. Unexpectedly, all transgenic strains generated with the same vector showed

identical hybridization patterns (Fig. 2a; Supplementary Fig. 3). This result would be compatible with transgene integration by homologous recombination, which would be theoretically possible, because the selectable marker gene is driven by expression signals derived from an endogenous *P. purpureum* gene (and, consequently, homologous recombination by a double crossover would result in knock out of the tubulin gene). An alternative explanation would be episomal maintenance of the transformation plasmid as autonomously replicating genetic element.

When uncut genomic DNA of transgenic *P. purpureum* stains was analyzed by Southern blotting and compared to undigested vector DNA, identical hybridization patterns were obtained corresponding to the characteristic open-circular and covalently closed circular plasmid conformations. This observation raised the possibility that the transformation vector is episomally maintained in the nucleus of the transgenic algal cells. Consistent with this interpretation, plasmid DNA linearized by restriction enzyme digestion produced a fragment of similar size as the genomic DNA of transformed algal strains cut with the same enzyme. Moreover, digestion of genomic DNA with enzymes that do not have cleavage sites in the transformation vector resulted in the same hybridization patterns as undigested genomic DNA (Fig. 2a).

Since the episome was not detectable by ethidium bromide staining of total genomic DNA (Fig. 2b), we sought to directly confirm its presence by plasmid rescue experiments in *E. coli*. To this end, bacteria were transformed with total DNA extracted from transformed algal strains followed by selection for ampicillin resistance, the bacterial selection marker present on the plasmid vector used for algal transformation. Plasmids were readily recovered in *E. coli*, and their molecular analysis revealed that they are identical to the transformation vector (Fig. 2c, d).

Unless maintained by selection, episomally replicating extrachromosomal elements are susceptible to occasional loss due to random segregation. To follow plasmid maintenance over time, transformed algal strains were grown in the absence of zeocin selection for multiple rounds of subculturing, and the percentage of plasmid-containing cells in the population was determined in each round. While no plasmid loss was detectable during three consecutive cultivation cycles, the plasmid abundance started to decline with the fourth round of subculturing (Fig. 2e). However, even after the sixth round, 60% of the cells still harbored the plasmid, indicating a remarkable stability even in the absence of selection for plasmid maintenance.

To investigate the basis for this high stability, we next determined the copy number of the plasmid in algal cells. Given the likely absence of a segregation mechanism for the foreign episome, high copy numbers could potentially explain the persistence of the introduced plasmid in the absence of selection. When plasmid copy numbers were measured, they were strikingly similar in all transgenic strains examined. Approximately 20 copies of the plasmid were estimated to be present per algal cell (Fig. 2f, g). This high value may explain the stability of the episome in the absence of selection (Fig. 2e) and likely contributes to the high transgene expression levels obtained (Fig. 1c–e). As expected, copy number and transgene expression level are correlated in that clones with lower copy numbers also show lower GFP accumulation (Supplementary Fig. 2).

**Plasmid replication in *P. purpureum* requires the bacterial origin**. The long-term persistence of the episome as circular plasmid in *P. purpureum* cells and the high copy number suggest very strongly that the plasmid is autonomously replicating as an extrachromosomal element. This conclusion raises the intriguing question how a bacterial plasmid can replicate as an extrachromosomal element in a eukaryotic system. We first

tested whether episome maintenance in the alga requires the circular conformation of the plasmid. To this end, vector pZL6 was linearized and used in large-scale transformation experiments. From more than 100 transformation experiments (i.e., shots with the particle gun), not a single zeocin-resistant clone was obtained (Fig. 3a; Table 1), strongly suggesting that maintenance in the alga requires the plasmid to be circular.

To dissect the sequence requirements for autonomous replication in *P. purpureum* cells, a series of deletion constructs and sequence replacement constructs were derived from transformation vector pZL6 (Fig. 3a; Table 1). Replacement of the expression elements controlling selectable marker gene expression affected neither transformation efficiency nor plasmid copy number in the alga (Table 1; Fig. 3b). Likewise, exchange of the marker gene for selection in *E. coli* (by replacing the ampicillin resistance gene *bla* with the kanamycin resistance gene *aphA1*) had no effect. By contrast, deletion of the replication origin (pMB1 ori) that is responsible for plasmid maintenance in *E. coli* abolished algal transformation completely. The corresponding deletion construct, pZL6(ΔpMB1), is unable to replicate in bacteria, and therefore needed to be generated by religation of large quantities of restriction enzyme-digested plasmid DNA in vitro. As a control, the same procedure was applied to generate a bacterial marker gene-free plasmid by removing the ampicillin resistance gene. While plasmid pZL6(ΔAmp^R) produced transgenic algal clones at high frequency, plasmid pZL6(ΔpMB1) did not result in any transformed clones, suggesting that the bacterial pMB1 ori is required for episomal maintenance of the plasmid in the alga.

We next wanted to determine whether activity in *P. purpureum* was a specific property of the pMB1 origin of replication or whether other bacterial replication origins would also be active in red algal cells. To this end, we replaced the pMB1 ori with the unrelated pSC101 ori that is derived from *E. coli* plasmid R6-5 (and is, for example, also present in cloning vector pBR322). While pMB1 replication and copy control relies on an antisense RNA mechanism[16], pSC101 utilizes the same host proteins for initiation of replication as the bacterial chromosome[17]. Interestingly, when tested in *P. purpureum*, the pSC101 ori also conferred autonomous replication of the plasmid in the alga and resulted in similar copy numbers as the pMB1 ori (Table 1; Fig. 3b). This finding indicates that the capacity to trigger autonomous replication in a eukaryotic system is not restricted to the pMB1 ori, but may be a common property of bacterial plasmids.

**Efficient co-transformation of *P. purpureum***. Having identified two plasmid species that can replicate in *P. purpureum* allowed us to conduct co-transformation and plasmid competition experiments. Due to the high number of plasmid molecules that are coated onto a single gold particle, the biolistic process introduces many copies of the transformation vector simultaneously into a single cell. We first mixed the two transformation vectors (pZL6 and pZL25; Fig. 3a; Table 1) in a 1:1 ratio, loaded the mixture onto gold particles and biolistically transformed algal cells. Unexpectedly, all analyzed transgenic algal clones turned out to be co-transformed and harbored both plasmids (Fig. 3c). However, the ratio of the two plasmids was somewhat variable between the transformed clones, indicating some stochasticity in the plasmid segregation process.

We also performed co-transformation experiments with uneven amounts of the two plasmids, by mixing the two vectors in a ratio of 1:9 or 9:1. As expected, this resulted in predominant presence of the more abundant transformation vector and reduced the co-transformation efficiency (Fig. 3c).

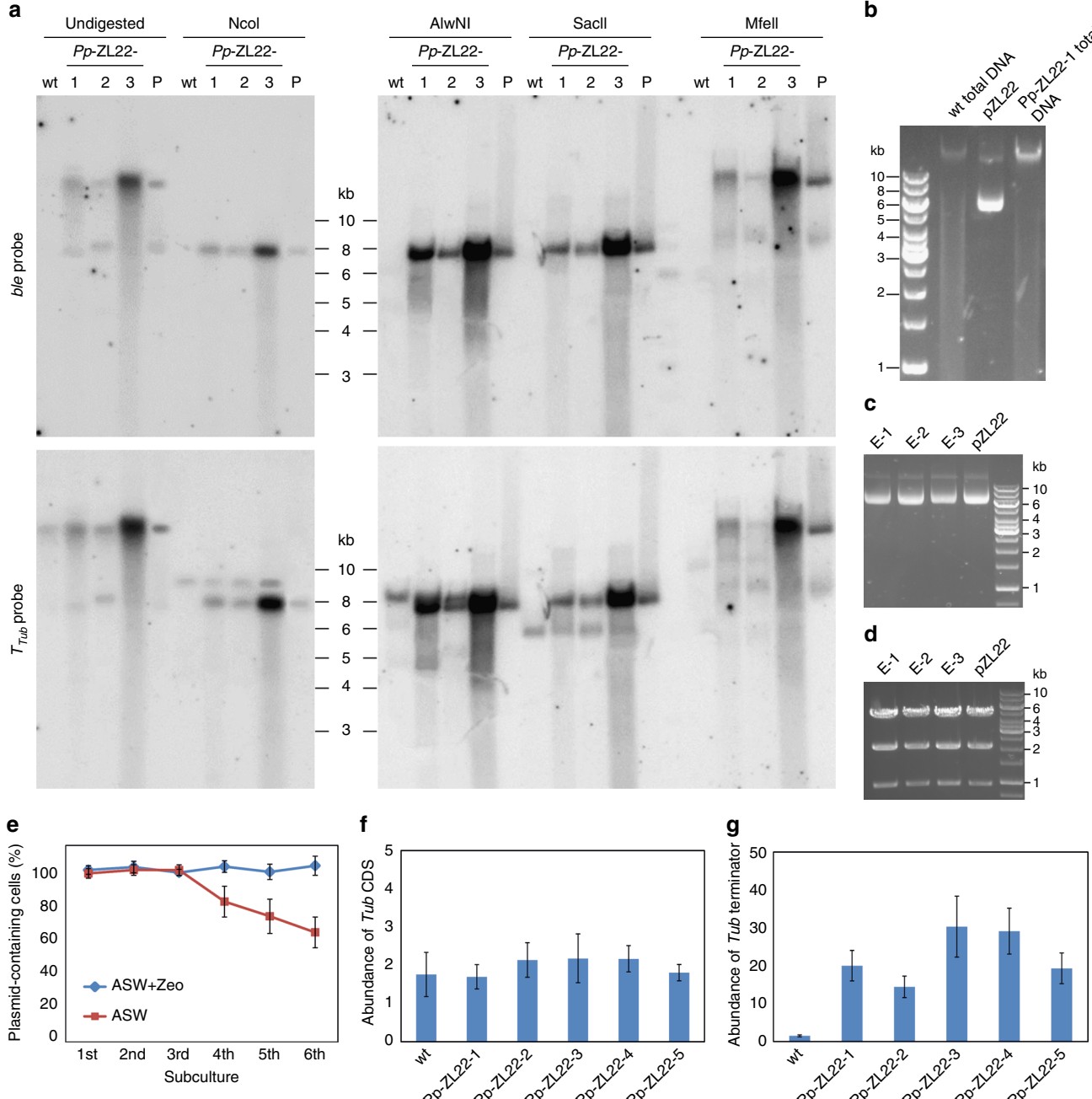

**Fig. 2** Transformed plasmids replicate as stable, circular, high-copy number episomes. **a** Southern blot of transgenic algal clones. DNA from the wild-type (wt) and three transgenic strains obtained with vector pZL22 was left undigested or cut with restriction enzymes that have a single recognition site in the vector. pZL22 isolated from *E. coli* (P) served as control. NcoI: methylation-insensitive enzyme, AlwNI: sensitive to dcm methylation, SacII: sensitive to CpG methylation, MfeII: methylation-insensitive (no site in pZL22). The blots were sequentially hybridized with the two probes (*ble* coding region, $T_{Tub}$ terminator). For Southern blots of transgenic strains generated with pZL6 and pZL19, see Supplementary Fig. 3. **b** Analysis of genomic DNA by gel electrophoresis. DNA from the wild-type strain, a transformed *Pp*-ZL22 strain, and the purified plasmid were separated in a 1% ethidium bromide-containing agarose gel. **c** Rescued plasmids isolated from *E. coli*. DNA of algae transformed with pZL22 was extracted 6 months after transformation (and continuous subculturing in selective medium) and transformed into bacteria. Plasmids were isolated from three antibiotic-resistant colonies (E-1, E-2, E-3) and analyzed by gel electrophoresis. Plasmid pZL22 was loaded as control. **d** Analysis of rescued plasmids by digestion with EcoRI. The rescued plasmids and the transformation vector pZL22 produce identical restriction patterns. **e** Plasmid loss assay with transgenic algal strains that were subcultured (by diluting the culture 1:1000) in artificial sea water (ASW) without selection for six successive cycles. After each cycle, aliquots were plated on agar-solidified ASW with and without zeocin. Plasmid loss was determined as the percentage of plasmid-containing cells (i.e., colonies on medium with zeocin: colonies on medium without antibiotic). **f, g** Determination of the copy number of the episomes relative to the algal nuclear genome. The abundance of the tubulin coding region (CDS; **f**) and the tubulin terminator (**g**) were determined by qPCR using DNA from the wild-type and five independent *Pp*-ZL22 transformants. The *EF1α* gene served as reference. Presence of the tubulin terminator in both the transgene and the tubulin gene in the genome allows copy number determination of the episome. Error bars: SD of the mean from three biological replicates. For qPCR efficiency of the primer pairs, see Supplementary Fig. 4

## Discussion

Red algae are an ecologically important group of algae. In marine ecosystems, they are major primary producers and also play a key role in building coral reefs. In addition, several red algal species are used as food (and represent a traditional part of Asian cuisine), and serve as raw material for the production of agar and other food additives. Despite the enormous ecological and economic importance of rhodophytes, molecular research on red algae is lagging behind research on plants and green algae.

In this work, we have addressed a bottleneck in red algal research by developing an efficient transformation system for the model red alga *P. purpureum*[5]. The system differs from all other transformation systems for algae and plants in that the transforming DNA does not integrate into the genome, but instead is maintained episomally as a high copy number plasmid. Natural plasmids are known to occur in some red algal strains[6,7], but are poorly characterized and their evolutionary origin is unknown. We have discovered here that bacterial plasmids can be stably maintained in red algal cells and two unrelated bacterial origins of replication confer autonomous replication. In view of this finding, a bacterial origin of naturally occurring plasmids in red algae (and their spread as parasitic elements[7]) will need to be considered. Sequencing of plasmid-harboring red algal strains should shed more light on the evolution of natural episomes in algae[7].

Recently, extrachromosomal elements derived from yeast sequences have been shown to be transferrable to diatom cells by bacterial conjugation[18]. These elements have similar copy numbers as the native chromosomes and their replication relies on eukaryotic sequences from yeast[18]. By contrast, plasmids in *P. purpureum* occur at high copy numbers of approximately 20 copies per cell (i.e., 10 times higher copy numbers than those of the chromosomes, considering that the sequenced strain of *P. purpureum* was likely diploid[5]). Moreover, plasmid maintenance in *P. purpureum* depends on bacterial origins of replication and thus represents an intriguing case of DNA exchange and genetic compatibility between two different domains of life (bacteria and eukaryotes). We, therefore, propose that the frequent presence of plasmids in red algae[6,7] might be explained by the compatibility of plasmid replication systems in bacteria with the DNA replication machinery operating in the red algal nucleus.

Formally, the possibility that the episomes replicate inside chloroplasts or mitochondria needed to be considered. However, maintenance of the episome in one of the organelles is exceedingly unlikely. First, marker and reporter genes are driven by eukaryotic expression signals that would not function inside chloroplasts or mitochondria[19], which, due to their endosymbiotic origin from bacteria, have retained prokaryotic gene expression machineries. Second, GFP lacking a transit peptide accumulated to high levels in the nucleus and the cytosol, whereas GFP tethered to a plastid transit peptide resulted in accumulation in the plastid compartment (Fig. 1c).

The mechanism of bacterial plasmid replication in the algal nucleus is currently unknown. Our data show that two different origins of replication can confer episomal maintenance in the nucleus (Fig. 3). Replication from the pMB1 origin depends on antisense RNA synthesis, and, in addition to RNA polymerase for transcription, requires DNA polymerase I and RNase H. By contrast, replication from the pSC101 origin requires the RepA protein (encoded in the ori region), but also the components involved in replication of the bacterial chromosome, including DNA polymerase I. While there is no additional RepA-like protein encoded in the *P. purpureum* genome[5], the alga has an unusually high number of genes encoding putative DNA polymerase I enzymes (5 *polA*-like genes: contig_3402.2, contig_3646.4, contig_3646.3, contig_2156.6, contig_2347.17; https://www.nature.com/articles/ncomms2931#supplementary-

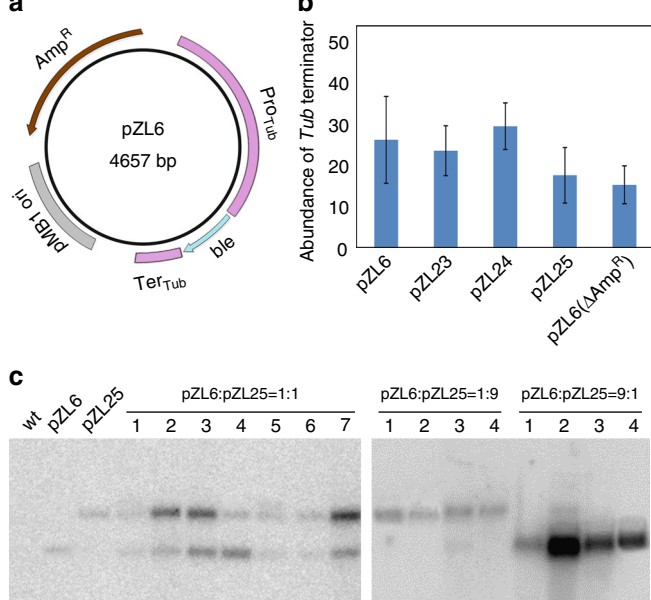

**Fig. 3** Bacterial replication origins confer episomal replication in *P. purpureum*. **a** Map of plasmid pZL6 with the main functional elements indicated. Amp$^R$: ampicillin resistance gene *bla*; Pro$_{Tub}$: tubulin gene promoter driving the selectable marker gene *ble* for algal transformation; *ble*: bleomycin (zeocin) resistance gene; Ter$_{Tub}$: terminator derived from the tubulin gene of *P. purpureum*; pMB1 ori: origin of replication of the plasmid in *E. coli*. **b** Measurement of episome copy numbers for the different constructs listed in Table 1. Copy numbers were determined by qPCR amplifying sequences from the tubulin terminator (cf. Fig. 2g). **c** DNA gel blot analysis of transgenic red algal strains generated by co-transformation with plasmids pZL6 and pZL25. Total cellular DNA from the wild-type strain (wt), the purified plasmids pZL6 and pZL25, and total DNA from red algal clones obtained by co-transformation with pZL6 and pZL25 were analyzed by Southern blotting. The gold particles used for biolistic transformation were coated with mixtures of plasmids pZL6 and pZL25 in three different ratios (1:1, 1:9, and 9:1). Note high-frequency co-transformation with the 1:1 mixture and more prevailing presence of the more abundant plasmid upon co-transformation with unequal plasmid ratios

information). Interestingly, three out of the five *polA*-like genes encode proteins that are not predicted to be targeted to the organelles (mitochondria and/or plastids). By contrast, the green alga *Chlamydomonas reinhardtii* has three putative *polA* genes, but their products are all predicted to localize to plastid or mitochondria. Whether or not the presence of additional, possibly nuclear, DNA polymerase I enzymes in *P. purpureum* facilitates the episomal maintenance of bacterial plasmids, will be interesting to determine.

Although nuclear transformation of *P. purpureum* has not been achieved before, there is a single published study reporting plastid genome transformation in this alga[20]. Reanalyzing the data and, especially, the DNA gel blots presented in this work, we identified no data that unambiguously support transgene integration into the plastid genome and would be inconsistent with episomal nuclear maintenance of the plastid transformation vector (Supplementary Fig. 5). We, therefore, believe that the previously reported plastid transformation of *P. purpureum* may, in fact, have been episomal nuclear transformation. In this context, it is also noteworthy that the study used an unconventional selectable marker gene that has not been successfully used for chloroplast

**Table 1 Identification of the plasmid region that confers episomal replication in *P. purpureum***

| Plasmid | Replication origin in *E. coli* | Selectable marker for *E. coli* | Promotor and terminator of *ble* gene | Transformation efficiency (cfu/µg) |
|---|---|---|---|---|
| pZL6 | pMB1 | Amp | Tubulin | 15 |
| pZL6 linearized | pMB1 | Amp | Tubulin | 0 |
| pZL6($\Delta$Amp$^R$) | pMB1 | – | Tubulin | 3 |
| pZL6($\Delta$pMB1) | – | Amp | Tubulin | 0 |
| pZL23 | pMB1 | Amp | Actin | 18 |
| pZL24 | pMB1 | Kan | Tubulin | 9 |
| pZL25 | pSC101 | Amp | Tubulin | 5 |

See "Methods" for experimental details

transformation in any other species[21], and, moreover, proposed transgene integration by a single crossover which normally cannot be observed in plastids (because, due to the high homologous recombination activity, only double crossovers can be detected).

The efficiency of co-transformation in *P. purpureum* was 100%, which, to our knowledge, is unparalleled in plant and algal transformation. We believe this to be due to the high copy number of the plasmid molecules present on a single gold particle and the high copy number of the plasmids in *P. purpureum*. GFP expressed in *P. purpureum* accumulated to extraordinarily high amounts of 5% of the total cellular protein, levels that are much higher than those normally attainable in transgenic algae or plants[14,15]. The high copy number of the episomes likely contributes to the high expression levels and makes *P. purpureum* an attractive future production system for recombinant proteins in algal biotechnology[4,22].

In sum, our work presented here (i) provides an efficient transformation system for the model red alga *P. purpureum*, (ii) suggests an explanation for the frequent presence of plasmid-like elements in red algal genomes, and (iii) may also have relevance to the high abundance of genes of bacterial origin in the nuclear genome of *P. purpureum*. Future research will need to test whether conditions can be identified that promote the integration of the episomes into the genome, and, in this way, reproduce HGT in the laboratory. Also, it would be interesting to develop transformation methods for other red algal species and assess the correlation between episomal transformation, residence time of plasmids in the nucleus, and the rate of HGT from bacteria.

## Methods
**Algal culture**. *P. purpureum* strain SAG 1380-1d (Culture Collection of Algae at Goettingen University, Germany; https://www.uni-goettingen.de/en/culture+collection+of+algae+%28sag%29/184982.html) was grown in liquid medium or on agar-solidified medium based on artificial sea water (ASW)[23] at 25 °C under continuous light (100 µE m$^{-2}$ s$^{-1}$). Liquid cultures were incubated under shaking at 125 rpm.

**Construction of transformation vectors**. For construction of a transformation vector based on the chloramphenicol acetyltransferase gene *cat*, the open reading frame of *cat* was cloned as an XhoI/BamHI fragment into a pUC18 backbone-based vector (pA7-GFP) that contained an expression cassette comprised of the CaMV-35S promoter and the *nos* terminator, generating plasmid pZL3. The zeocin resistance gene *ble* from *S. hindustanus* was amplified by PCR and cloned as an XhoI/BamHI fragment into the same vector, producing plasmid pZL4. The α-tubulin gene promoter and terminator of *P. purpureum* were amplified by PCR using total DNA as template. The tubulin gene promoter, the *cat* coding region, and the tubulin terminator were cloned into the AccI, BamHI, and KpnI sites of pUC18, resulting in a tubulin expression cassette driving *cat* expression (vector pZL5; Fig. 1b). The *cat* coding sequence of pZL5 was then replaced by *ble* (inserted into the BamHI site), yielding vector pZL6 (Fig. 1b). The *GFP* coding region was amplified by PCR using pA7-GFP as template. For construction of pZL19, PCR products covering *ble* and *GFP* were inserted into the BamHI site of pZL6 by recombination cloning using the In-Fusion® HD Cloning Kit (Clontech, Mountain View, CA, USA). The promoter and terminator of the actin gene of *P. purpureum* and the transit peptide of the gene for the plastid ATP synthase γ-subunit (*ATPC*) was amplified by PCR using red algal total DNA as template. The actin terminator, actin promoter, and *ATPC* transit peptide were successively cloned into the XbaI/

KpnI, PstI/XbaI, and XbaI sites of pUC18, resulting in an actin expression cassette driving expression of a GFP targeted to the plastid. This cassette was subcloned into the EcoRI site of pZL6, generating transformation vector pZL22 (Fig. 1b). The fragment comprising the actin promoter, the *ble* coding region, and the actin terminator was cloned into the AccI/KpnI sites of pUC18 by recombination cloning to produce pZL23. The ampicillin-resistant gene of pZL6 was replaced with the kanamycin-resistant gene *aphA1* (from *E. coli* transposon Tn903) by excision with AhdI and SphI, generating vector pZL24. The pMB1 origin of replication of plasmid pZL6 was replaced by the pSC101 origin of replication as an AhdI/PciI fragment, generating vector pZL25. The sequences of all primers used in this study are listed in Supplementary Table 1.

**Transformation of *P. purpureum***. For biolistic transformation of *P. purpureum*, $10^8$ cells were harvested at a density of $10^6$ cells mL$^{-1}$ and bombarded on a plate with agar-solidified ASW medium supplemented with the appropriate antibiotic as selection agent. To this end, 0.5 g gold particles (0.6 µm diameter; Bio-Rad, Munich, Germany) were coated with 1 µg plasmid DNA and shot at the cell lawn from a distance of 9 cm using 1350 psi rupture disks. Bombardments were conducted with a DuPont PDS-1000/He biolistic gun (Bio-Rad). After bombardment, the plate was incubated in the dark for 24 h and then transferred to continuous light. Resistant colonies appeared after approximately 10 days. Zeocin-resistant algae were selected with 25 mg L$^{-1}$ zeocin.

**DNA isolation, Southern blot analysis, and real-time PCR**. For DNA isolation, algal cells were harvested by centrifugation ($5000 \times g$, 5 min), washed twice with ASW, resuspended in 60 °C extraction buffer (2% cetyltrimethyl ammonium bromide, 100 mM Tris–HCl pH 8.0, 20 mM EDTA pH 8.0, 1.4 M NaCl, 2% polyvinylpyrrolidone 40, 2% β-mercaptoethanol), and incubated for 30 min. The lysate was extracted twice with phenol/chloroform/isoamyl alcohol (25:24:1) and then treated with RNase A for 20 min at 37 °C, followed by extraction with chloroform/isoamyl alcohol (24:1). DNA was precipitated with 0.7 volume isopropanol followed by washing of the pellet with 70% ethanol. Finally, the dried DNA pellet was dissolved in water.

For Southern blot analysis, total DNA were digested with appropriate restriction enzymes, separated by electrophoresis in 1% agarose gels, and transferred onto Hybond XL nylon membranes (GE Healthcare, Buckinghamshire, UK) by capillary blotting. For preparation of hybridization probes, the *ble* coding sequence and the terminator of the tubulin gene were amplified by PCR (Supplementary Table 1), purified by agarose gel electrophoresis and labeled with [α-$^{32}$P]dCTP by random priming (Megaprime$^{TM}$ DNA Labeling System; GE Healthcare). Hybridization was performed at 65 °C using standard protocols.

Real-time quantitative PCR assays were performed using the LightCycler 480 Real-Time PCR System (Roche Applied Science, Penzberg, Germany) and ABsolute SYBR Green ROX mix (Thermo Scientific, Waltham, MA, USA). Total cellular DNA was used as PCR template. The abundances of the coding region and the terminator of the tubulin gene were normalized to the gene for elongation factor 1 alpha (*EF1α*). Three biological replicates were analyzed with three technical replicates each. The $2^{-\Delta\Delta CT}$ (cycle threshold) method was used to determine relative DNA levels. Statistical evaluation was performed by analysis of variance with Microsoft Office Excel 2010.

**Protein extraction and immunoblot analysis**. TSP was extracted by resuspending red algal cell pellets in 200 µL lysis buffer (50 mM HEPES/KOH pH 7.5, 10 mM KAc, 5 mM MgAc, 1 mM EDTA, 1 mM DTT, 1× protease inhibitor cocktail cOmplete [Roche]), followed by disruption of cells by sonication (Sonifier®, W-250 D; G. Heinemann Ultraschall und Labortechnik, Schwäbisch Gmünd, Germany) at 10% amplitude for 15 s on ice. Proteins were quantified using the Bradford assay (RotiQuant, Roth, Karlsruhe, Germany).

For immunoblotting, protein samples were denatured at 95 °C for 3 min in sample buffer (62.5 mM Tris–HCl pH 6.8, 17.5% glycerol, 2.1% SDS, 100 mM DTT, 0.015% bromophenol blue), separated in denaturing 15% SDS–PAA gels and transferred onto polyvinylidene difluoride membranes (Hybond$^{TM}$ P; GE Healthcare) using a Trans-Blot electrophoretic transfer cell (Bio-Rad) and a

standard Tris-glycine transfer buffer (25 mM Tris/HCl, 192 mM glycine, pH 8.3). Recombinant GFP (Roche) was used as standard for semiquantitative assessment of protein accumulation. Immunochemical protein detection was performed using a 1:5000 dilution of monoclonal anti-GFP primary antibody (Clontech, Mountain View, CA; Cat. 632381) and a 1:10,000 dilution of anti-mouse HRP-conjugated secondary antibody (Agrisera, Vännäs, Sweden; AS11 1772). Hybridization signals were visualized by the ECL[TM] Prime detection system (GE Healthcare).

**In-gel fluorescence assays and microscopic methods**. TSP was isolated under non-denaturing conditions using the standard extraction protocol described above with the following modifications[13]. DTT was omitted from the lysis buffer, the final SDS concentration was decreased to 1% in the sample buffer, the protein samples were directly loaded onto the PAA gel without prior incubation at 95 °C, and the electrophoretic separation was performed at 4 °C. For in-gel detection of fluorescence, gels were scanned with a Typhoon[TM] Trio[+] scanner (GE Healthcare)[13]. Detection of GFP in living cells was done with a confocal laser-scanning microscope (TCS SP5; Leica, Wetzlar, Germany) using an argon laser for excitation (488 nm), a 500–510 nm filter for detection of GFP fluorescence, and a 630–720 nm filter for detection of chlorophyll fluorescence.

**Plasmid loss assay**. The assay was based upon six successive subcultures of a 1:1000 dilution of algal cultures in ASW without antibiotic. From each subculture, aliquots were spread on ASW agar plates with and without zeocin (25 mg L$^{-1}$). Plasmid loss was determined as the ratio between the number of colonies on the plate with zeocin and the number of colonies on the plate without zeocin.

**Identification of plasmid region conferring replication in the alga**. By removing individual functional elements, a series of deletion constructs was derived from plasmid pZL6 and tested in algal transformation experiments. Linearization of pZL6 was done with the restriction enzyme NdeI. The ampicillin resistance gene was removed from pZL6 by digestion with FspI and religation with T4 DNA ligase to generate the circular vector pZL6(ΔAmp$^R$). The pMB1 ori sequence was cut out with AseI followed by religation, resulting in plasmid pZL6(ΔpMB1). The tubulin gene promoter and terminator were replaced by the promoter and terminator from the actin gene of *P. purpureum* generating plasmid pZL23. The ampicillin resistance gene was replaced by a kanamycin resistance gene (*aphA1*) producing pZL24. The pMB1 ori was exchanged with the pSC101 ori generating pZL25. Note that only the linearized plasmid pZL6 and the plasmid lacking the bacterial origin of replication do not result in transgenic algal clones.

**Data availability**. The data supporting the findings of this study are available within the paper and its Supplementary Information files.

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

## Acknowledgements

We thank Dr. Stephanie Ruf and Pierre Endries for help with transformation experiments, Dr. Rouhollah Barahimipour for help with the in-gel fluorescence assays, and Dr. Daniel Karcher (all MPI-MP) for helpful discussion. We are grateful to Professor Debashish Bhattacharya (Rutgers University, NJ) for providing algal strains and cultivation protocols. This work was supported by a grant from the European Research Council (ERC) under the European Union's Horizon 2020 research and innovation program (ERC-ADG-2014; grant agreement 669982) to R.B., and by the Max Planck Society.

## Author contributions

Z.L. performed all experiments and analyzed data. R.B. conceived of the project, analyzed data, and wrote the manuscript.

## Additional information

**Competing interests:** The authors declare no competing interests.

