## [Peer Review File · Nature Communications]

Reviewers' comments:

Reviewer #1 (Remarks to the Author):

The manuscript on transformation of *Porphyridium* by Li and Bock is very well written, technically strong, and of high importance to the field of lateral gene transfer in algae and other unicellular eukaryotes. This area remains controversial because mechanisms of foreign gene integration are not apparent and the lack of genetic tools for groups such as red algae does not allow the direct testing of hypotheses. The future goal of eliciting integration of plasmid-borne DNA into the nuclear genome of *Porphyridium* is, in my opinion, the most exciting prospect offered by this work. For these reasons I strongly favor publications of this manuscript in *Nat Commun* and have only a few minor comments to make.

Intriguingly, Fig. 7 in Lee et al. (2016) that provides a model of plasmid spread in red algae was prescient, predicting (unknowingly) the current results, and should be specifically cited.

A very surprising aspect of the results was the replication of plasmid DNA in the red algal nucleus. This suggests that plasmid encoded genes have a long residence time in the red algal nucleus to facilitate LGT. In that way, plasmids are critical among prokaryotes being considered as a pool of extrachromosomal DNA shared among populations. Given that a lot is known about plastid replication mechanisms, it would be very useful if the authors could provide a model for how this occurs in *Porphyridium* (i.e., initiation, elongation, and termination). Do these plasmids require a Rep initiator protein borne on this DNA (and transcribed) or do bacterial derived proteins (or candidates) exist in the red algal nuclear genome? Which mechanism of DNA replication is occurring (theta, rolling circle)? This may be trivial but will help readers gain insights into how red algae maintain plasmids, that ultimately may play a role in providing adaptive functions.

Reviewer #2 (Remarks to the Author):

In the present manuscript, the authors report the development of an efficient genetic transformation system for the model rhodophyte *Porphyridium purpureum*, in which following transformation, the transgenic DNA is maintained as extrachromosomal high copy number plasmid. Interestingly, the data provide evidence that the bacterial origin of DNA replication confers replication in the algal nucleus. Probably due to the high copy number plasmid, expression of transgenes can be up to 5 % of the total soluble protein. In conclusion this is an important step forward towards efficient transgene expression in *Porphyridium purpureum* with strong implication for mechanistic studies and biotechnological applications. It further implies that the occurrence of extrachromosomal plasmids in red algae may also have contributed to the high rate of horizontal gene transfer from prokaryotic source organisms to the nuclear genome of *Porphyridium purpureum*.

Further comments.

1. This work establishes the development of an efficient genetic transformation system for *Porphyridium purpureum* on the basis of an extrachromosomal high copy number plasmid. However, does this hold true for other red algae model systems? This seems to be also important in respect to their conclusion in regard to horizontal gene transfer from prokaryotic source organisms to the nuclear genome of red algae. It is also important in regard to a wider applicability.

2. The authors suggest that the stable chloroplast transformation in *Porphyridium purpureum* as

shown in reference 20, is due to episomal nuclear transformation. Although not necessary for their conclusions, it would be of importance if they could prove their case.

3. As the high copy number plasmid is decreased in the absence of antibiotic selection, is transgene expression altered correspondingly? In Fig.S2 transformants 3 and 4 show lower protein expression as compared to transformants 1 and 2 after 6 month of transformation, is their plasmid copy number different?

4. A similar question arises from the plasmid completion experiments; does the transgene expression alter in regard to plasmid ratio correspondingly?

5. Fig. S2, is this six-month after transformation and subsequently in the absence of selection? Please indicate in the figure legend. The phycoerythrin loading control is not perfectly equally loaded.

Response to Reviewers (comments: black; response: red)

Reviewer #1 (Remarks to the Author):

The manuscript on transformation of *Porphyridium* by Li and Bock is very well written, technically strong, and of high importance to the field of lateral gene transfer in algae and other unicellular eukaryotes. This area remains controversial because mechanisms of foreign gene integration are not apparent and the lack of genetic tools for groups such as red algae does not allow the direct testing of hypotheses. The future goal of eliciting integration of plasmid-borne DNA into the nuclear genome of *Porphyridium* is, in my opinion, the most exciting prospect offered by this work. For these reasons I strongly favor publications of this manuscript in *Nat Commun* and have only a few minor comments to make.

Intriguingly, Fig. 7 in Lee et al. (2016) that provides a model of plasmid spread in red algae was prescient, predicting (unknowingly) the current results, and should be specifically cited.

The model of parasitic plasmid spread proposed by Lee et al. (2016) is now specifically mentioned (p.12).

A very surprising aspect of the results was the replication of plasmid DNA in the red algal nucleus. This suggests that plasmid encoded genes have a long residence time in the red algal nucleus to facilitate LGT. In that way, plasmids are critical among prokaryotes being considered as a pool of extrachromosomal DNA shared among populations. Given that a lot is known about plastid replication mechanisms, it would be very useful if the authors could provide a model for how this occurs in *Porphyridium* (i.e., initiation, elongation, and termination). Do these plasmids require a Rep initiator protein borne on this DNA (and transcribed) or do bacterial derived proteins (or candidates) exist in the red algal nuclear genome? Which mechanism of DNA replication is occurring (theta, rolling circle)? This may be trivial but will help readers gain insights into how red algae maintain plasmids, that ultimately may play a role in providing adaptive functions.

As suggested, we have addressed possible mechanisms of plasmid replication in the algal nucleus by bioinformatics methods. While there is no RepA protein encoded in the algal genome, we found an unusually large number of homologues of the bacterial DNA polymerase I. Interestingly, 3 out of the 5 putative DNA polymerase I enzymes are predicted to NOT be targeted to plastids or mitochondria and, thus, could operate in the nucleus (where they potentially could participate in plasmid replication). We describe these findings on p.13 of the revised manuscript and highlight the functional analysis of these putative DNA polymerase I genes as an interesting topic of future research into the mechanism of plasmid maintenance.

Reviewer #2 (Remarks to the Author):

In the present manuscript, the authors report the development of an efficient genetic transformation system for the model rhodophyte *Porphyridium purpureum*, in which following transformation, the transgenic DNA is maintained as extrachromosomal high copy number plasmid. Interestingly, the data provide evidence that the bacterial origin of DNA replication confers replication in the algal nucleus. Probably due to the high copy number plasmid, expression of transgenes can be up to 5 % of the total soluble protein. In conclusion this is an important step forward towards efficient transgene expression in *Porphyridium purpureum* with strong implication for mechanistic studies and biotechnological applications. It further implies that the occurrence of extrachromosomal plasmids in red algae may also have contributed to the high rate of horizontal gene transfer from prokaryotic source organisms to the nuclear genome of *Porphyridium purpureum*.

Further comments.

1. This work establishes the development of an efficient genetic transformation system for *Porphyridium purpureum* on the basis of an extrachromosomal high copy number plasmid. However, does this hold true for other red algae model systems? This seems to be also important in respect to their conclusion in regard to horizontal gene transfer from prokaryotic source organisms to the nuclear genome of red algae. It is also important in regard to a wider applicability.

It is currently unknown, how widespread episomal transformation may be in (red) algae. This is because very few algal species are currently transformable, and we know only of a single other red algal species for which a workable transformation protocol has been published. Testing other red algae would require at least a few years of research (to optimize the culture conditions, identify antibiotic sensitivities, develop markers and conduct transformation experiments). We have added a short paragraph to the Discussion to highlight the need to develop similar transgenic tools for other species and to assess the correlation between episomal transformation and the rate of horizontal gene transfer (p.14).

2. The authors suggest that the stable chloroplast transformation in *Porphyridium purpureum* as shown in reference 20, is due to episomal nuclear transformation. Although not necessary for their conclusions, it would be of importance if they could prove their case.

As requested, we now provide the full evidence for the erroneous conclusions in the published report on chloroplast transformation. These include a complete reanalysis of both the published PCR dataset and the Southern blot data. This analysis is presented in

Supplementary Fig. 5 of the revised manuscript. We now also explain (in the text; p.13) another problem with the published study: The erroneous data interpretation is based on integration by a single crossover, which is highly unlikely to be correct (single crossover intermediates are known to be undetectable in plastid transformation).

3. As the high copy number plasmid is decreased in the absence of antibiotic selection, is transgene expression altered correspondingly? In Fig.S2 transformants 3 and 4 show lower protein expression as compared to transformants 1 and 2 after 6 month of transformation, is their plasmid copy number different?

4. A similar question arises from the plasmid completion experiments; does the transgene expression alter in regard to plasmid ratio correspondingly?

Both comments relate to the question whether transgene expression correlates with plasmid copy number. We have addressed this point experimentally by determining copy numbers in all samples shown in Fig. 1e and Fig. S2a. As expected, there is a strong correlation between copy number and GFP expression levels. The data are shown in Fig. S2b, explained in the figure legend (and referred to in the main text).

5. Fig. S2, is this six-month after transformation and subsequently in the absence of selection? Please indicate in the figure legend. The phycoerythrin loading control is not perfectly equally loaded.

Information added to the figure legend. The phycoerythrin is not a loading control, it just happens to be detectable by our imaging method. The gels were loaded based on equal protein amounts, as stated in the legend to Fig. 1e.

REVIEWERS' COMMENTS:

Reviewer #2 (Remarks to the Author):

The manuscript has further improved and all points raised were adequately addressed.